# GraphTyper2 enables population-scale genotyping of structural variation using pangenome graphs

Hannes P. Eggertsson [1,2]*, Snaedis Kristmundsdottir[1,3], Doruk Beyter [1], Hakon Jonsson [1], Astros Skuladottir[1], Marteinn T. Hardarson[1], Daniel F. Gudbjartsson [1,2], Kari Stefansson[1,4], Bjarni V. Halldorsson [1,3]* & Pall Melsted[1,2]*

Analysis of sequence diversity in the human genome is fundamental for genetic studies. Structural variants (SVs) are frequently omitted in sequence analysis studies, although each has a relatively large impact on the genome. Here, we present GraphTyper2, which uses pangenome graphs to genotype SVs and small variants using short-reads. Comparison to the syndip benchmark dataset shows that our SV genotyping is sensitive and variant segregation in families demonstrates the accuracy of our approach. We demonstrate that incorporating public assembly data into our pipeline greatly improves sensitivity, particularly for large insertions. We validate 6,812 SVs on average per genome using long-read data of 41 Icelanders. We show that GraphTyper2 can simultaneously genotype tens of thousands of whole-genomes by characterizing 60 million small variants and half a million SVs in 49,962 Icelanders, including 80 thousand SVs with high-confidence.

[1] deCODE genetics/Amgen Inc., Sturlugata 8, Reykjavik, Iceland. [2] School of Engineering and Natural Sciences, University of Iceland, Reykjavik, Iceland. [3] School of Science and Engineering, Reykjavik University, Reykjavik, Iceland. [4] Faculty of Medicine, School of Health Sciences, University of Iceland, Reykjavik, Iceland. *email: hannese@decode.is; bjarnih@decode.is; pmelsted@decode.is

Characterization of sequence variants in the human genome has greatly improved[1–5] with lower sequencing costs and improvements in sequencing technologies. Particularly small sequence variants (SNVs, indels, and small complex variants), which modify fewer than 50 nucleotides, are usually detected by finding discordances in short-read alignments compared with a reference genome[6–8]. In a previous publication, we presented GraphTyper[6], a method for population genotyping small sequence variants using pangenome graphs[9–11] (or genome graphs). Pangenome graphs are a graphical representation of multiple genomes. Graphs can be utilized to extend the linear reference genome such that they are aware of sequence variants. Each path in such a graph encodes a potential haplotype. In summary, GraphTyper constructs its graph from a reference genome and a set of sequence variants in variant-call format (VCF). GraphTyper then extracts reference aligned reads from a small genomic region, locally realigns them to a pangenome graph and, concurrently, genotypes variants in the graph. We showed that our method refines alignments near variation and can jointly call tens of thousands of samples.

Sequence variants that modify 50 base pairs or more are known as structural variants (SVs). While SVs are only a small portion of all sequence variation, analyses suggest that they have a high impact on gene expression[12] and they have been implicated in many rare diseases[13–15]. Recent studies suggest that each individual has on average over 27 thousand SVs[16], but some of them can only be discovered using specific library preparation or sequencing technologies, such as long-read sequencing.

Short-read whole-genome sequencing (WGS) technologies are widely used in population-scale genotyping. They are readily available and have low error rates. However, due to their read length limitations, SVs need to be discovered from read assemblies, split-read alignments, read alignment coverage, read-pair insert sizes or other indirect inferences. These SV discovery methods have lower sensitivity and specificity than methods aimed at smaller variants. Further, breakpoints of the detected SVs are often imprecise and the SV sequence is often partially characterized[4].

Population-scale genotyping refers to when samples from the same population are genotyped together, either one at a time or jointly. Joint calling is typically favored for population-scale genotyping as it generates a set of genotype calls, which are comparable across the samples in the population and can be used directly in genome-wide association studies. In the widely-used Genome Analysis ToolKit HaplotypeCaller[7], genotyping small variation in a population is performed by joint calling from intermediate files (gVCF), which contain support (or lack of support) for a variant at every position of the genome. The data are then combined across all samples to generate a variation map for the population. While this approach is effective for joint calling single-nucleotide variants (SNVs), it may have difficulties calling other variants. Calling SVs is in particular problematic, as the exact SV boundaries are often imprecise and thus may be represented differently between samples.

The problem of variant calling can be split into discovery and genotyping. In the discovery step, potential variation sites are detected and in the genotyping step genotypes are called at those sites. A number of methods exist for both discovering SVs using short-read data[17–20], however, the sensitivity of these methods is limited by the read length[16]. Multiple high-quality assemblies constructed with long-read data have been made publicly available[21–25]. These assemblies provide a great resource of sequence variants[26,27], some of which are difficult to discover using short-reads.

Here we present a second version of GraphTyper (GraphTyper2) that enables efficient encoding of structural variation into the pangenome graph and genotyping of those variants. Our method can now jointly genotype both small variants and SVs with short-reads at a population scale. GraphTyper supports most simple types of SVs, including deletions, insertions, duplications, and inversions. While our method discovers small variants, it relies on external resources for SV discovery, such as SVs derived from long-read assemblies[27] and short-read SV discovery methods[17–20].

## Results

**SV genotyping workflow.** The main data structure in GraphTyper is a directed acyclic graph (DAG), where a path in the DAG represents a possible haplotype. One of the possible haplotypes is the reference sequence, which GraphTyper requires as input. Variants are alternative sequences compared with the reference. SVs can be encoded in the DAG and can coexist in the graph with small variants. We define a breakpoint to be the location where an SV either diverges from or converges to the reference and a breakpoint sequence to be the SV sequence near the breakpoint (Fig. 1a). Two breakpoints are typically added for each SV, representing the start and end locations of the SV with respect to the reference. It is also possible that only one of these locations was discovered, in which case the SV is represented by a single breakpoint.

To limit the size of the graph, GraphTyper inserts only the breakpoint sequences (up to 152 bp – determined by the short-read length) into the graph. This limits compute time and allows robust SV genotyping across SV lengths, as the mapping is not biased toward larger SVs. Another advantage is that the alternative SV sequence is often only partially characterized. At each SV breakpoint, the reference allele and the alternative allele are represented by nodes in the graph containing the reference sequence and the SV breakpoint sequence, respectively. GraphTyper realigns all sequence reads of a genomic region, including clipped and unaligned reads with an aligned mate read in the region, to the graph structure and genotypes the variants encoded in that graph. GraphTyper genotypes SVs in a graph along with previously discovered SNPs and indels (Fig. 1b).

In our pipeline for genotyping SNPs and indels, we partition the genome into 50 kbp regions (by default) and genotype each region separately. We use larger graphs to genotype SVs (default 1.2 Mbp and overlap by 200 kbp) such that the breakpoints of each SV are typically in the same graph. Of the SVs generated in the 1000G project[4], only 36 (0.052%) would have had breakpoints on different graphs if they were binned using our scheme. For those larger SVs, we can expect less accurate results from GraphTyper.

Prior to genotyping, we extract all reads aligned to the region and realigned them to the pangenome graph. Misalignment near SV breakpoints often leads to false positive variant calls (Supplementary Fig. 1), a problem that can be alleviated by realigning the reads onto a variation-aware data structure[6,11].

GraphTyper has two models for genotyping SVs, one is based on read realignments to the SV breakpoints and the other is based on alignment coverage ("Methods"). Briefly, each breakpoint is genotyped by comparing the number of reads aligning to the reference and alternative sequences. Further, deletions and duplications are also genotyped from decrease and increase in alignment coverage, respectively. The aggregated genotype call is made by selecting the call among all genotype models that has the highest genotyping quality. We evaluated the genotyping models using the svclassify's[28] deletion set and determined that the aggregated calls have the highest recall than the other two models (Supplementary Methods).

The graph construction, indexing, and alignment otherwise follows what we described previously[6].

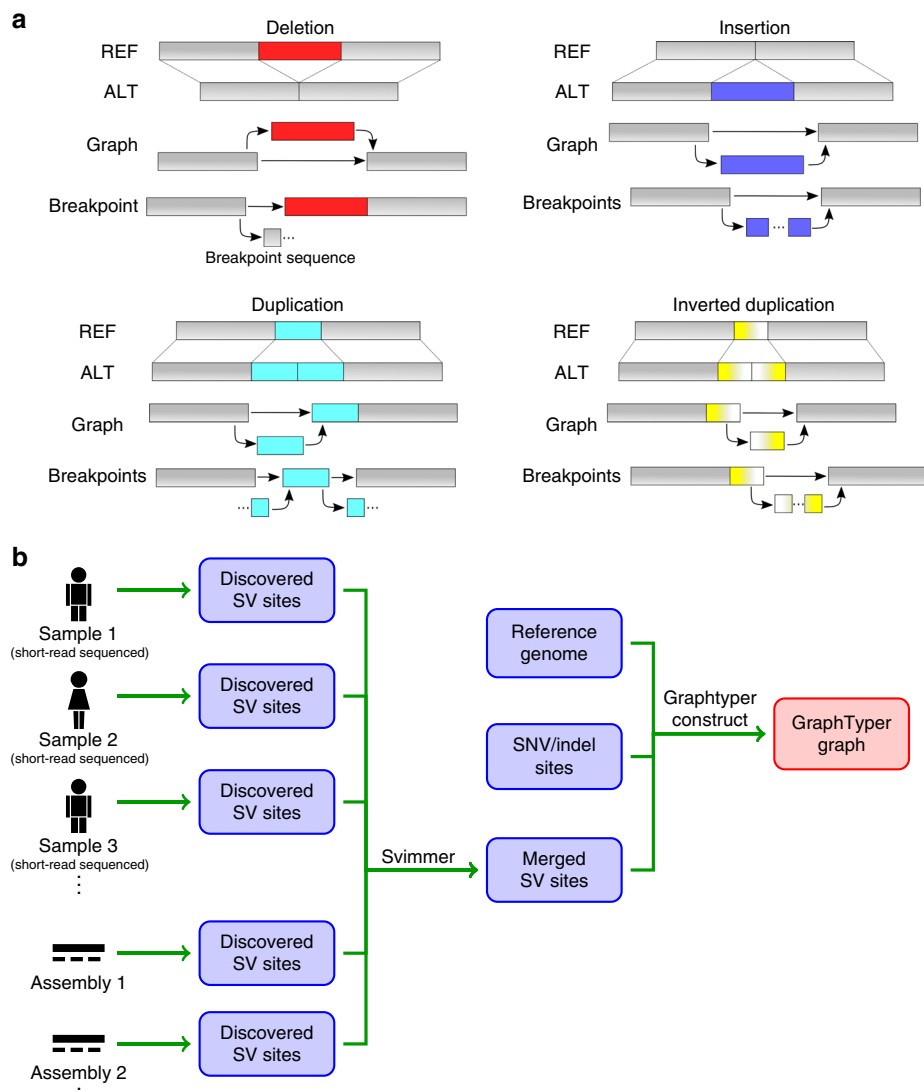

**Fig. 1** Overview of data structure and workflow. **a** Example structural variants and their encoding in an acyclic graph structure. **b** Workflow for constructing a GraphTyper graph with SNPs, indels and SVs. SVs are detected from each sample independently and then merged across all the samples, such that SV sites of the same type and similar position and size are reported only once. SNPs and indels that are given as input into the graph construction can be detected using GraphTyper or obtained from a database.

**Single sample genotyping performance**. We evaluated Graph-typer's SV genotyping performance using the recently published syndip benchmark dataset[27]. The dataset contains sequence variants of a synthetic-diploid genome, which was derived from de novo long-read assemblies of two homozygous cell lines, CHM1 and CHM13. The dataset has 18,630 autosomal SVs in high-confidence regions with respect to the hs37d5 reference genome, however, we expect to identify much fewer SVs with short-reads. Whole-genome sequence data that was sequenced from an even mixture of DNA from both cell lines is available ("Methods").

From the short-reads, we called SVs using three state-of-the-art methods: Delly[17], Manta[18], and smoove (SV discovery using Lumpy[19] and genotyping using SVTyper[29]). We also discovered SVs from public de novo long-read assemblies of six ethnically diverse individuals (Supplementary Table 1) using a pipeline derived from the one used in syndip[27]. By incorporating these assemblies, the goal is to capture many of the common SVs that are difficult to discover using short-reads only. We ran GraphTyper with three different input SV sets: SVs discovered using Manta (Manta + GraphTyper), SVs discovered with both

Manta and from four assemblies where the CHM1 and CHM13 assemblies were excluded (Manta + UA + GraphTyper), and SVs discovered with both Manta and from all six assemblies (Manta + AA + GraphTyper). Manta + AA + GraphTyper is expected to have a higher recall than Manta + UA + GraphTyper and the other methods as it contains SVs discovered from the assemblies that were used to create the truth set.

In our comparison we considered an SV to be recalled if it was of the current SV type and had breakpoints within a selected precision threshold in base pairs. For deletions, Manta + GraphTyper obtained a higher sensitivity (40.0–46.0%) than the other methods (5.1–43.4%) at every tested breakpoint precision threshold (Fig. 2a). Manta + GraphTyper is able to obtain higher sensitivity than Manta since GraphTyper considers Manta SVs which did not pass Manta's quality metrics. By incorporating the four unrelated assemblies and all six assemblies, the sensitivity was substantially increased up to 53.1–63.5% and 67.6–72.9%, respectively. Manta and Manta + GraphTyper had the lowest false discovery rate (FDR) at strict breakpoint thresholds (up to 17 bp), however, smoove had lower FDR at more lenient

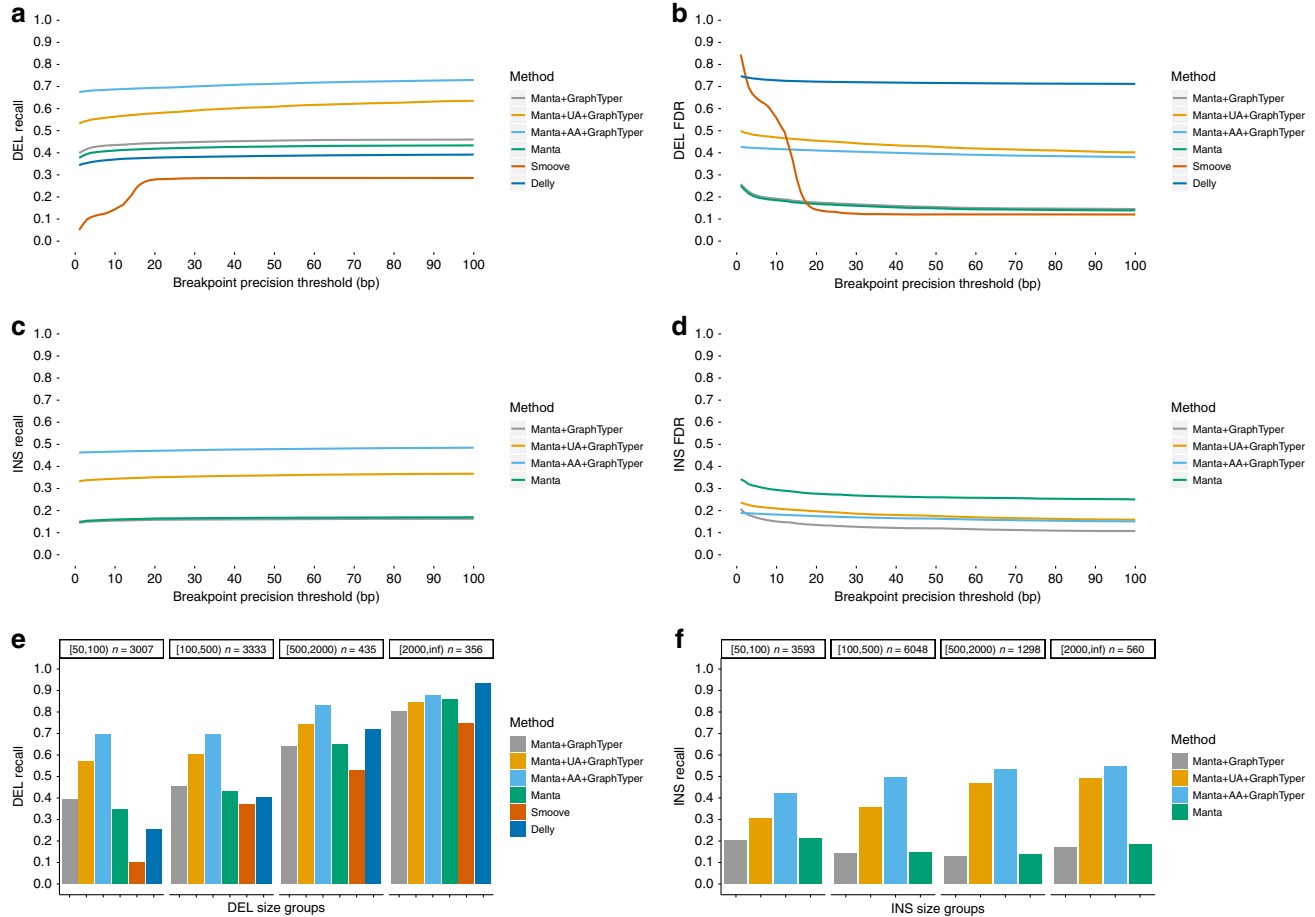

**Fig. 2** Comparisons to SVs in the syndip dataset. The breakpoint precision threshold is the maximum number of base-pairs we allowed at both breakpoints for an SV to be considered recalled. **a** Deletion recall comparison between SV genotyping methods. **b** Deletion false discovery rate comparison. The Manta and Manta + GraphTyper lines are overlapping. **c** Insertion recall comparison. Delly and smoove were not evaluated since they are not designed to discover all types of insertions. The Manta and Manta + GraphTyper lines are overlapping. **d** Insertion false discover rate comparison. **e** Deletion recall by deletion size with a breakpoint precision threshold of 50 bp. **f** Insertion recall by insertion size with a breakpoint precision threshold of 50 bp.

thresholds (Fig. 2b). We saw that Manta + AA + GraphTyper had considerably more false positives than Manta + GraphTyper, however, most of them (87%) did not pass our high-confidence genotype call filter. We decided not to enforce that filter by default though, as it would remove 43% of the true positives as well.

For insertions, we did not evaluate Delly and smoove since they do not call all types of insertions (only duplications). Manta and Manta + GraphTyper had a low and comparable insertion sensitivity (11.3–13.6%) (Fig. 2c). However, the FDR of Manta + GraphTyper (10.8–20.8%) was lower than Manta's FDR (25.1–34.2%) (Fig. 2d), indicating that GraphTyper does not genotype many of the false positive Manta SVs. Manta + UA + GraphTyper and Manta + AA + GraphTyper obtained a substantially higher sensitivity (33.3–36.7% and 46.3–48.5%, respectively). The results suggest that utilizing long-read assemblies, even public assemblies of unrelated samples, can greatly benefit short-read SV genotyping.

We analyzed how SV size, as it is reported in the truth set, might influence recall. Interestingly, we saw that all methods had the lowest sensitivity when genotyping deletions of sizes 50–99 bp and highest when genotyping deletions of size 2000 bp or larger (Fig. 2e). When assessing insertions, we saw that Manta and Manta + GraphTyper had the highest recall when genotyping 50–99 bp insertions (20.9% and 20.3%, respectively) compared with larger insertions, while Manta + UA + GraphTyper and

Manta + AA + GraphTyper had the highest recall when genotyping insertions of 2000 bp or larger (49.1% and 54.6%, respectively). The results show that the utilization of long-read de novo assemblies improve recall across all deletion and insertion sizes. We observed the largest improvements on large insertions, where recall was up to threefold higher than the best method that did not utilize the assemblies.

**Genotyping four Icelandic families.** In our comparison to a public dataset, our evaluation was restricted to a single genome. Since our method is intended for population-scale genotyping, we further evaluated it by genotyping chromosome 20 of 56 individuals in four Icelandic families (Fig. 3a). The families consist of 8 parents and 48 offsprings: two families have 10 offsprings, one family 11, and one 17 offsprings. We merged SVs discovered by Manta and genotyped them using GraphTyper ("Methods") and compared the results to joint calling all 56 individuals using Manta, Delly, and smoove. In this study, we did not include any SVs from long-read assemblies since we wanted to assess GraphTyper's genotyping performance on Manta variants only and see how it compares with the original Manta calls.

We identified Mendelian inheritance errors in the 48 parent-offspring trios. The high-confidence Manta + GraphTyper genotypes had a very low inheritance error rate (0.27%), more than tenfold lower than the high-confidence Manta genotypes had

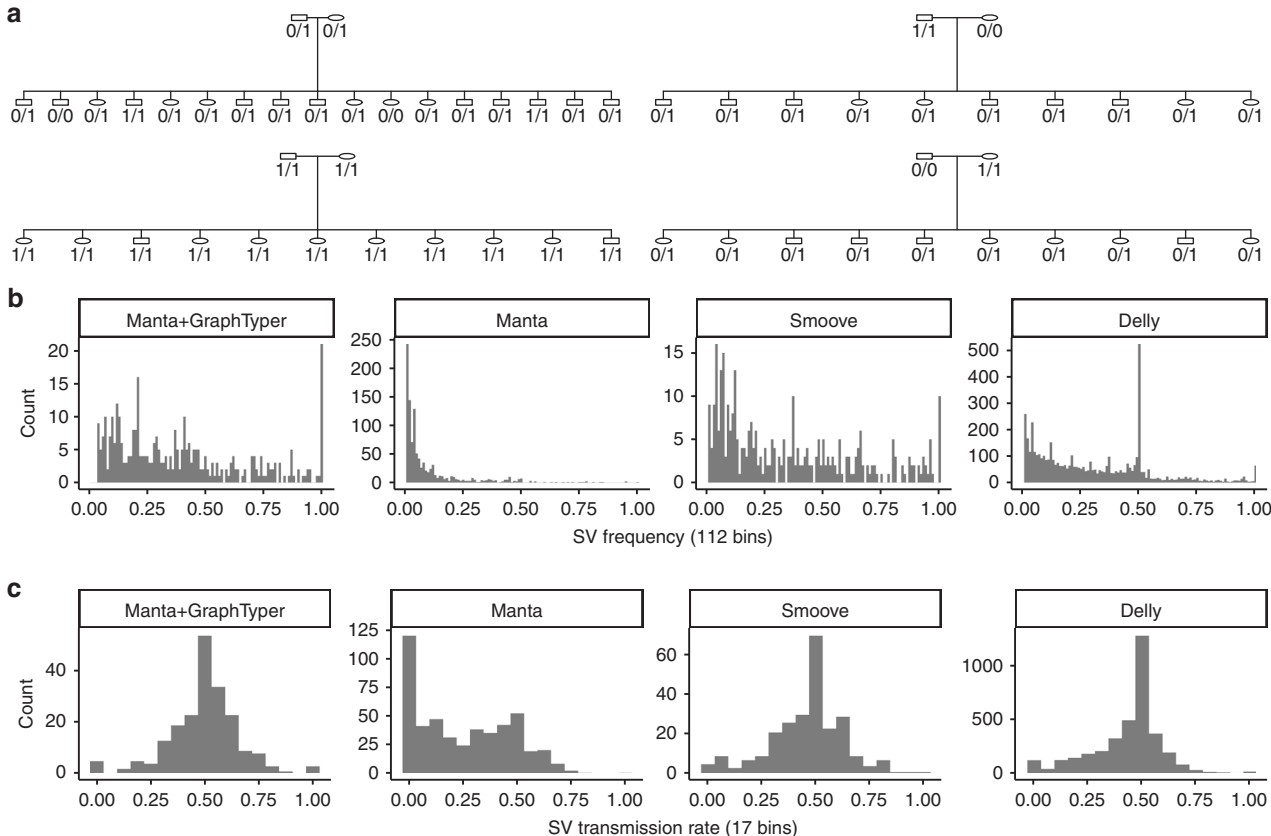

**Fig. 3** High-confidence SV genotypes in four Icelandic families. **a** Family tree of the four families. Shown are genotypes of a 313 bp deletion starting at chr20:19,080,772 (GRCh38). **b** Frequency distribution of SVs called on chromosome 20. There are 112 bins, the number of chromosomes in the callset. **c** The allele transmission rate of an SV from parent to offspring. For germline variants, the distribution is expected to be symmetric around 50%.

(Supplementary Table 2). Compared with the other methods, Manta + GraphTyper also genotypes the offsprings with a call distribution much closer to the expected values based on the parent genotypes. All individuals in the large families have at least ten close relatives, therefore we would expect that almost every SV is carried by multiple individuals. In the Manta joint calls we surprisingly saw that 22.2% of the SVs had only one carrier (Fig. 3b), while Manta + GraphTyper did not genotype any SVs in only one carrier.

Next, we measured the transmission rate of SV alleles in the families ("Methods"). Assuming Mendelian inheritance, germline alleles transmit from parent to offspring with 50% probability. We thus expect that high-quality variants to have transmission rate distribution symmetric around 50%. The distribution of Manta + GraphTyper calls was close to being symmetric around the 50% (Fig. 3c), while the Manta joint calls were heavily biased toward lower transmission rates, indicating erroneous genotyping.

Based on the above observations, we conclude that Manta is useful for discovering SVs but underestimates their frequencies when joint calling. By merging Manta variants from multiple samples and using them as input for GraphTyper, we can alleviate that problem.

**Long-read validation**. While variant segregation in families can reveal genotyping inaccuracies, they do not verify that the SVs are of the correct size and type. To tackle this, we sequenced the genomes of 41 Icelanders using long-read sequencing ("Methods") and compared the short-read SV calls to those of Sniffles'[30]. The two methods are orthogonal as GraphTyper only uses Illumina short-reads (median coverage 38.3×, range 25.1× to 164.7×)

while Sniffles uses Oxford Nanopore long-reads (median coverage 13.4×, range 8.3× to 33.9×). We expect to detect more true SVs in long-reads than in short-reads, although with a lower breakpoint accuracy. We required two long-reads to support an SV of the same type (deletion or insertion) for it to be considered validated and used a maximum breakpoint threshold of 50 bp (Supplementary Methods).

In the 41 Icelanders, we genotyped 25,790 high-confidence SVs using the Manta + AA + GraphTyper workflow. On average per genome, 10,938 high-confidence were genotyped and, thereof, we validated 6812 SVs (62.3%) with the long-reads (Supplementary Table 3). We validated more deletions (3572 on average) than insertions and duplications (3240 on average), which is consistent with previous short-read SV studies[4,31]. Using a more lenient maximum breakpoint threshold of 100 bp and 200 bp, we validate 67.7% and 71.9% of the high-confidence SVs, respectively, which indicates that many SVs are not validated because they may have inaccurate breakpoint positions in either the short-read or long-read SV calls.

**Population-scale SV genotyping**. We assessed the scalability of our method by genotyping a large cohort of 49,962 whole-genome sequenced (median coverage 36.9×, range 17.8× to 307.3×) Icelandic genomes ("Methods"). We ran the experiment before we had incorporated the SV discovery from long-read assemblies into our workflow and thus we only used SVs discovered by Manta SVs. In total, 543,939 SVs were discovered in the population. These SVs were added to the SNP and indel graphs that were previously created with GraphTyper. Subsequently, we genotyped all samples using GraphTyper. After filtering SVs with no non-reference genotypes, we retained 486,158

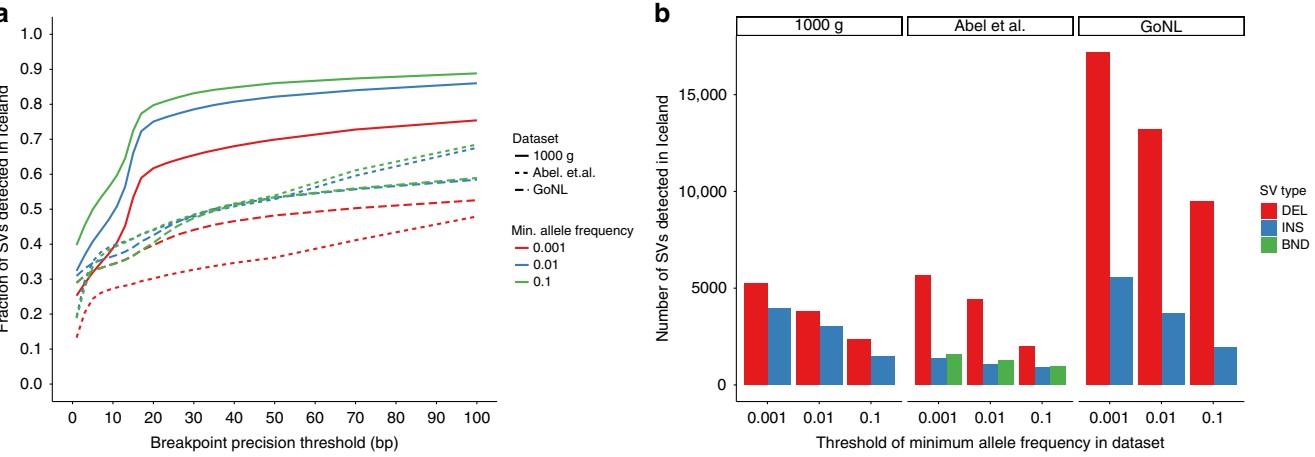

**Fig. 4** Overlap of previously published SV datasets and SVs we find in Iceland. **a** Fraction of SVs in an external SV dataset that are also found in Iceland. **b** Distribution of the number of insertions, deletions, and breakends of an external dataset that is found in Iceland. Maximum distance threshold used was 50 bp.

SVs and thereof, we considered 79,318 as high-confidence SVs (Supplementary Methods).

We analyzed how many SVs in previously published datasets[2,4,31] overlapped with our SVs ("Methods") (Fig. 4). As expected, we found a greater overlap with common variants and more with deletions than insertions. In the same genotyping run, GraphTyper genotyped 59.5 million small variants: 44.3 million SNPs, 4.0 million indels and 11.2 million other small variants. Merging the SVs required 2,223 CPU hours of compute time and genotyping using GraphTyper took 4.15 million CPU hours or 83 CPU hours per sample on average. Our results show that GraphTyper is a practical solution for high-quality genotyping across many different variant classes, both with few and many samples.

## Discussion

We have extended our previously published variant caller to enable SV genotyping in large-scale datasets. With our extension, GraphTyper can be used to genotype variants across many variant classes without any variant size restrictions. Our experiments suggest that GraphTyper is sensitive in comparison to other widely-used structural variant callers. We also show that it can be applied to genotype 50 thousand genomes, which is to our knowledge the largest WGS-based SV set in terms of number of genomes. We believe our approach of realigning reads to a variation-aware graph is important for decreasing mapping bias toward the reference genome and improving genotype quality. Our analysis emphasizes the importance of jointly genotyping SVs as it has clear advantages, resulting both in greater sensitivity and lower inheritance error rate.

GraphTyper does not completely fulfill the promises of pan-genomes. Its graph data structure is limited as it cannot represent every type of sequence variation, for example nested variants in SVs. Further, our method extracts reads based on their mapped position to the linear reference genome and therefore it has reference bias. A previous study suggested that using vg's graph alignment prior to GraphTyper could marginally increase the number of true positive SNP and indel calls[9]. Applying a similar strategy would be possible for SV graphs, but first we would require high-quality SV population graphs. Unfortunately, these graphs do not exist yet. This may change with improvements in sequencing technologies and graph-based methods for representing and calling SVs, some of which are in active development[32,33].

Our method has several other limitations. For instance, GraphTyper's breakpoint model genotypes SVs based on realignments to an SV sequence, but some SVs can have homologous sequences at the breakpoint, resulting in ambiguous alignments. Further improvements are needed to be able to genotype those SVs reliably, for example by incorporating read-pair information. Additionally, we designed our current SV merging algorithm to be simple and scale to 50,000 individuals. We think it should be optimized in the future, for example by taking into account the error profile of the discovery methods and the evolutionary processes of mutations.

We believe that with our method, SVs can be more easily incorporated into large sequencing studies. Here we have demonstrated that our method is sensitive, genotypes SVs accurately, and can be robustly applied to very large WGS datasets. We showed that by incorporating public long-read data we can greatly improve SV sensitivity, in particular of large insertions. We expect further improvements as long-read sequencing becomes more available. Our next goal is to further investigate our Icelandic population SV results and find phenotypic implications of the genotyped SVs. Further, our method has a very low error rate compared with previous methods and thus we have paved a way for high-quality de novo SVs analysis to further study the origin and mechanics of SVs.

## Methods

**Icelandic DNA data**. The Icelandic samples were whole-genome sequenced at deCODE Genetics using Illumina GAIIx, HiSeq, HiSeqX, and NovaSeq sequencing machines, and sequences were aligned to the human reference genome[34–36] (GRCh38) using BWA-MEM[37].

The 41 Icelandic samples with long-read data were also sequenced using Oxford Nanopore Technologies sequencing machines and basecalled using Albacore (version 2.1.3). The reads were mapped using minimap2[38] (version 2.14). The average alignment coverage was calculated using samtools'[8] depth (version 1.9). SVs with at least two supporting reads were discovered using Sniffles[30] (version 1.0.10).

DNA was isolated from both blood and buccal samples. All participating subjects signed informed consent. The personal identities of the participants and biological samples were encrypted by a third-party system approved and monitored by the Data Protection Authority. The National Bioethics Committee and the Data Protection Authority in Iceland approved these studies.

**Merging of SV sites across samples**. Many SV discovery methods do not joint-call SVs but discover SVs on each sample independently, or on a small number of samples simultaneously. In each sample, the same SV may be reported slightly differently due to imprecise breakpoint resolution. To avoid populating the graph structures with multiple versions of the same SV we created a method for merging SV sites from many single samples VCFs into a single VCF file. Our SV merging

method is called svimmer and is similar to the one used in SURVIVOR[39], however, we could not use SURVIVOR since it replaced the original INFO field of the VCF when merging. Many values in the INFO field are required to be able to represent the SV correctly. In svimmer, the INFO field from the original VCF is retained in the final output. Also, svimmer ignores the samples' genotype information to reduce compute time and memory, as only SV site information is needed for GraphTyper's graph construction.

Svimmer groups all SVs that are of the same type, have a size difference within 100 bp (default value), and where both begin and end position are within 200 bp (default value) of each other. If an SV that fulfills these criteria with any other SV in a group then it is merged into the group. To prohibit any group from getting extremely large, we disallowed an SV to be added in a group if either its begin or end position is further than 10,000 bp apart from any SV of that group.

When all SVs have been merged into groups, we find the most common pair of begin and end positions and select an SV with those positions to be a representative for the group in the final output. While merging SVs discovered by Manta we merged all reported SVs, including those that did not pass in Manta's filter.

**Comparison to syndip**. For comparing SV callers we used the syndip dataset for the 37d5 reference genome. The dataset also contains small variants and thus we extracted a truth SV set by comparing the size difference of the reference allele compared with each alternative allele at the variant site, if the length difference was 50 bp or greater we retained the alternative allele. Multi-allelic records were first decomposed and all records were normalized. We assessed only the SVs inside the high-confidence regions from syndip, however, we expanded the high-confidence regions by 25 bp before extracting SVs from the caller sets to account for imprecise SV breakpoints. The commands for the SV calling and the comparisons are shown in the Supplementary Methods.

**Relative genotype likelihoods**. The genotyping process selects the two most likely haplotypes (or paths) in the graph based on the read data we observed. All SVs are genotyped independently of each other by comparing how many reads support a given SV breakpoint allele sequence compared with the reference allele sequence (Supplementary Fig. 2). We consider a read to support an allele if its best graph alignment overlaps the allele. The VCF input can have multi-allelic SV sites and they are represented as such in the graph. A read might have equally good alignments to more than one allele if a sequence is shared between the alleles. We handle those cases by saying that all those alleles are considered supported. We do not expect to frequently observe two SV events occurring at the same position in the same sample and therefore GraphTyper genotypes multi-allelic SV sites as two or more biallelic sites (reference allele vs. alternative allele).

Given an SV site with alleles $x$ and $y$, let $G_{xy}$ be the unphased SV genotype of a sample and let $R$ be the multiset of the sample's reads that have a graph alignment that overlaps the SV breakpoint. For biallelic variants, the unphased SV genotype can only be $G_{00}$, $G_{01}$, or $G_{11}$. Here, allele 0 denotes the reference allele and allele 1 denotes the alternative allele.

The relative genotype likelihood of each of those genotypes are

$$L\left(R|G_{xy}\right) = \prod_{r \in R} L(r|G_{xy}), \qquad (1)$$

where the relative likelihood of observing read $r$ given the genotype is

$$L\left(r|G_{xy}\right) = \begin{cases} 1, & \text{if both alleles } x \text{ and } y \text{ are supported by read } r \\ 1/2, & \text{if exactly one of } x \text{ and } y \text{ are supported by read } r, \\ \varepsilon_r, & \text{if neither alleles } x \text{ nor } y \text{ are supported by read } r \end{cases} \qquad (2)$$

where we arbitrarily chose that $\varepsilon_r$ is $1/2^8$ if the read is paired and its mate mapped onto the graph and $1/2^4$ otherwise. We consider an allele to be supported by a read if the graph alignment of the reads overlaps that allele. GraphTyper calls the genotype that has the highest relative likelihood for each sample.

We created a genotyping model to estimate genotypes of SV deletions and duplications (including inverted duplications) based on the drop and increase of alignment coverage in the graph, respectively. Each graph alignment is aligned back to the reference haplotype and the alignment coverage is stored at each reference base-pair. To measure the coverage drop or increase, we look-up the alignment coverage every 20 bp and determine the median coverage in two 1000 bp windows flanking the SV, $c_{out}$, and median coverage inside the SV, $c_{in}$. We selected 1000 bp since it gave us a good estimate of the alignment coverage in a window while being unlikely to overlap other SVs.

For deletions, we say that the coverage decrease, $\max(0, c_{out} - c_{in})$, is the number of reads supporting the deletion while $c_{in}$ is the number of reads supporting the reference (i.e. no deletion). We calculate relative genotype likelihoods using Eq. 2 with $\varepsilon = 1/2^4$. For duplications the genotype likelihoods are calculated similarly but with coverage increase instead of decrease.

**Parent-offspring trio transmission rate**. We defined the SV transmission rate to be the rate at which an SV is transmitted from a heterozygous parent to his/her offspring. The rate can be measured for every SV that has at least one heterozygous parent since then his/her SV allele is expected to transmit to the offspring with a probability of 50%, assuming that the allele is present in the germline. The observed mean transmission rate is often below 50% though, due to false positive variants, somatic variants, reference bias and more.

**Reporting summary**. Further information on research design is available in the Nature Research Reporting Summary linked to this article.

## Data availability
Access to the raw Icelandic sequence data, that support the findings of this study, is available on request from KS. The data are not publicly available because of Icelandic state law. Illumina reads for the synthetic-diploid CHM1/CHM13 sample is in the European Nucleotide Archive under accession PRJEB13208. The syndip dataset[27] was obtained from https://github.com/lh3/CHM-eval (v0.5). The long-read de novo assemblies were acquired from the links provided in Supplementary Table 1. The Illumina for NA12878, NA12891 and NA12892 were obtained from the Platinum Genome project[40] and the deletion truth set for NA12878 was obtained from the Supplementary Information of svclassify's article[28].

## Code availability
GraphTyper is available at https://github.com/DecodeGenetics/graphtyper (v2.0-beta, GNU GPLv3 license). Svimmer, our SV merging software, is available at https://github.com/DecodeGenetics/svimmer (v0.1, GNU GPLv3 license). We obtained smoove (v0.2.3) from https://github.com/brentp/smoove.

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

## Acknowledgements

We are grateful to our colleagues from deCODE genetics / Amgen Inc. for their contributions. We also wish to thank all research participants who provided a biological sample to deCODE genetics.

## Author contributions

H.P.E. implemented the GraphTyper software. H.P.E. and S.K. implemented the SV merging software. H.P.E., S.K., P.M. and B.V.H. designed the algorithms. H.P.E., P.M., B.V.H., K.S. designed the experiments. S.K., H.J., and M.T.H. contributed software for the study. H.P.E. wrote the initial version of the manuscript and S.K., D.B., H.J., A.S., M.T.H., D.F.G., P.M., B.V.H., and K.S. contributed to the subsequent versions. All authors reviewed and approved the final version of the manuscript.

## Competing interests

All authors are employees of deCODE Genetics/Amgen, Inc.
