## [Peer Review File · Nature Communications]

Reviewers' comments:

Reviewer #1 (Remarks to the Author):

In this manuscript, the authors extend their GraphTyper tool to handle linear structural variants (insertions and deletions). This generalization is straightforward given the graphical models that they use to support their genotyping approach. The particular encoding is somewhat unusual, as instead of building a graph from the full SV allele, the authors add structural variants to the graph by adding only sequence around their breakpoints. It is also somewhat unusual in that it appears (at least based on figure 1) to be reference biased. The reference sequence is retained in the case of deletions, but the insertion sequence is not in the case of non-reference insertions. I am curious if this might affect performance for insertions and deletions. Could the authors perhaps demonstrate that their approach does not cause reference (or deletion) bias in the genotyping of indels?

As in their previous work, their approach here is not based on a whole genome alignment of reads to a pangenome graph, but a local realignment of reads that are anchored to a particular genomic region to a chunk of the pangenome at that locus. This limitation is not as severe as it might seem, and they showed previously that this expedient approach improves genotyping accuracy at small variants, in particular indels. (My own experience confirms this, as I developed a similar technique that was applied in the 1000 Genomes Project to yield high quality indel genotype likelihoods, but was not published outside of the supplementary material of the final project paper.)

Although its use of pangenome graphs is interesting, GraphTyper is also impressive in terms of its joint genotyping capability, and can handle tens of thousands of samples efficiently. Here, the authors build on the graph based features of GraphTyper and its joint genotyping scalability to produce a new method capable of genotyping SVs in large population cohorts. The scale that they achieve is impressive, and represents the largest combined genotyping effort for SVs that I am aware of. As such, the significance of this work cannot be questioned. However, I would like that the authors spend some more time validating their method, and I will try to provide some suggestions about other methods that they might be able to use.

To build up to their final 50k genome result, the authors first validate the approach by using a truth set established with a previous paper (svclassify) on NA12878 and comparing their performance to other SV genotyping tools (figure 2). They combine this with a study of apparent SV transmission in four large families from the Icelandic cohort, to demonstrate that the Manta+GraphTyper pipeline produces accurate SV calls that are likely to obey expected allele transmission patterns. They then examine the rate of re-call of SVs from a number of sources in the 50k sample Icelandic dataset, which provides some cross-validation as to the accuracy of the method.

To better understand the performance characteristics of their method, I suggest that the authors attempt to build and genotype a set of SVs from the "syndip" synthetic diploid benchmark (<https://www.nature.com/articles/s41592-018-0054-7>). This context provides two PacBio assemblies of fully homozygous cell lines and a lab-mixed Illumina data set combining the two lines. I suggest that the authors use SVs found by aligning these assemblies to the linear reference to build their graph, and then attempt to re-genotype them using the mixed Illumina data. This kind of evaluation would be the most generic possible given current data, and would allow the authors to describe the performance of their method for any kind of SV found between these two assemblies and the reference genome. I understand that this is a significant ask, and must clarify that the paper in its current form presents a significant advance. I do think that the authors should consider adding another simulation study to evaluate the performance of their method where the truth can be (nearly) known for certain. This particular design, or a modulation of it (such as focusing on a subset of manually curated SVs) might be a simple way to tackle it with available data.

I have a few minor comments about the text.

First, on line 56 of page 4, the authors describe the gVCF merging approach for joint genotyping, and note that it breaks for SVs. I would argue that there is ample evidence that this approach breaks even in the case of small indels and complex or clustered variants, due to the effects of differential allele alignment and low coverage over complex alleles. A joint genotyping approach like GraphTyper has substantial advantages in the case of complex alleles, and the authors are more than right to point out that it matters even more for SVs.

Second, on line 59 of page 4, the authors describe pangenome graphs as extensions of the linear reference that include sequence variants. I contest this, in that I regard "pangenome graphs" as the representation of whole genome alignments of many genomes. Thus there is no particular reference genome to add variation to, but many that are equally represented. The particular case in which we have a linear reference genome that is generalized to include variation is of course useful, and equivalent to a VCF or DAG version of the pangenome. The authors' particular implementation does behave as they describe, and I think they should clarify here (as they do later) that they are working on a particular subset of the general idea as it is applied in practice, and not lay claim to the generic concept.

In their discussion, on page 12, line 204-205, they note that GraphTyper can be applied to the full structural variant spectrum. I find this slightly at odds with the fact that their method doesn't directly handle some kinds of SVs, like CNVs (amplifications), inversions, and translocations. Perhaps the authors should qualify this statement, as they do in other parts of the document.

It's a minor issue that will surely be caught later, but I noticed that a paper I coauthored is listed twice in the works cited.

I thank the authors for their excellent work on this topic,

Erik Garrison

Reviewer #2 (Remarks to the Author):

The paper proposes a new method (GraphTyper 2) to genotype structural variations (SVs) using pangenome graphs. The approach is a continuation of the authors' previous work (GraphTyper) on genotyping smaller variations (SNVs). The method consists of predicting putative SVs using a standard reference-based approach on a set of short reads on a population sample, adding start and end points (breakpoints) of these SVs to a pangenome graph consisting of SNVs that have been predicted using GraphTyper on the same data, and finally using re-aligned reads in the vicinity of breakpoints to do the final prediction of genotypes.

The method is compared against competing approaches that do not use pangenome graphs. A slight sensitivity increase is observed in experiments on a standard public dataset (100G, family trio, high-confidence deletions). This is at the cost of precision, so that the approach is competitive in F1 score only on a small range of parameter values tested (predicted breakpoint location accuracy).

While the public dataset is limited in this context, authors proceed to analysing in-house data, consisting of short and long read data on 56 and 41 Icelanders, respectively, from 4 families. The high number of offspring in the chosen family trees gives a valuable way to evaluate precision of genotype calls: Same inherited genotype should appear in more than one individual. In this experiment, the new approach appeared to provide much more sensible genotype calls than competing approach. Another analysis on family trios gave further evidence, since the transmission rate predictions of the new method were distributed around 50%, while the competing method was far from this. Long read data was used as a further evidence on the correctness of the short read calls: 67.7% of calls were supported by long reads.

As a summary, while the improvements on public dataset are rather modest, the in-house experiments complement well. As far as I know, this kind of analysis on large family cohort is novel. I find this analysis a bigger contribution than the proposed method itself, which consists of

somewhat ad hoc steps like greedy clustering of close-by breakpoints, huge number of fixed parameters (breakpoint size, coverage cutoffs, etc.). Some of these are unavoidable, but I would appreciate more rigorous approaches to the subproblems involved for repeatability and adjustability to different situations.

The success on in-house data raises some concerns. Why was only Manta used as a comparison? Why not Manta+BayesTyper, Delly, Lumpy+SVTyper, as with the public dataset?

Authors provide an implementation of Graphtyper2. The software works as expected: Can be installed easily and runs without problems on small dataset. On data on a complete chromosome a standard desktop is not sufficient, so such analyses need to be run on servers equipped with sufficient main memory.

Some more specific comments follow.

Authors do not formally define a breakpoint, but introduce an implicit definition in lines 87-88, which is the intuitive meaning of breakpoint. However, the word breakpoint and breakpoint sequence are used in ways that appear to mean something else, without much clarification. Figure 1.a and Supplementary Figure 2 are supposed to help, but they add little or no insight.

Some excerpts:

87-88: "Two breakpoints are typically added for each SV, representing the start and end locations of the SV (...)"

92-93: "GraphTyper inserts only the breakpoint sequences (up to 152 bp (...))"

So, if breakpoints are locations, what exactly are breakpoint sequences ?

Moreover lines 111-113 look almost like a circular definition:

111-113: "Briefly, breakpoints are genotyped by counting the number of reads aligning through the different paths of a breakpoint. Each breakpoint is genotyped separately for variants with two breakpoints."

Figure 1.a should clarify this, but it doesn't help much.

One could hope that the method sections would provide more precise definition, but that was not the case, as in section "Relative genotype likelihoods" we still see some ambiguity, e.g in

390 "(...) comparing how many reads support a given SV breakpoint compared to the reference allele" we get the impression that the SV breakpoint is actually the actual SV allele. However, immediately after:

391-392 "We consider a read to support an allele if its best graph alignment is in a path that overlaps the allele"

Again, supplementary Figure does not add much insight here.

Now, after line 399 "SV breakpoint" appears to mean the actual breakpoint as originally stated.

Then a second comment (minor) about the experiments;

188-192 "As before, we merged SVs discovered by Manta which resulted in 543,939 SVs discovered in the population. These SVs were added to the SNP and indel graphs that were previously created with GraphTyper. Subsequently, we genotyped all samples using GraphTyper which resulted in 486,158 SVs that were called in at least one sample."

What is the reason that the result after using GraphTyper is a reduced number of SVs ? (maybe it is obvious, but still a comment will be welcomed).

Moreover, this would appear to be at odds with Figure 2 (although it is a different dataset), in which Manta+GraphTyper has almost the same precision as Manta alone, and the actual

improvement of using Manta+GraphTyper (against just Manta) is in the sensitivity.

Last, the merging of nearby breakpoints appears a bit ad hoc. Have you considered using a dynamic programming approach for segmenting the breakpoint locations? Or using some clustering tool that can also take the breakpoint sizes into account (defining a distance between two breakpoints as a function of distance and size difference)?

Reviewer #3 (Remarks to the Author):

Eggertsson et al described GraphTyper2, a graph-based short-read genotyper for both small variants and structural variations (SVs). It is the successor of GraphTyper published in Nature Genetics two years ago. The major improvement in this new version is the ability to call SVs.

GraphTyper2 takes a list of potential SVs, incorporates them into a sequence graph and genotypes them in a set of samples. In the manuscript, the authors ran GraphTyper2 on SV calls generated by the Manta SV caller. To some extent, GraphTyper2 refines Manta calls, takes the advantage of multi-sample information and achieves higher accuracy, as is shown in authors' evaluation. The idea behind GraphTyper2 is sound and the algorithm is relatively straightforward. I believe it can be applied to a variety of large cohorts for SV calling which is currently missing in most studies.

Critical observations are as follows:

* In the manuscript, the authors generated SV candidates with Manta from short reads. Its sensitivity is largely limited by short reads. On the other hand, I see the key advantage of genotyping is that we can use SV candidates found by other technologies. I believe the authors can achieve much higher sensitivity by incorporating long-read SVs into the graph.

Here is a proposal. 1) Download a dozen or so long-read assemblies described in PMID:30661756. 2) Map these assemblies to the human reference genome using the pipeline described here: <https://github.com/lh3/CHM-eval/tree/master/dip-call> (the page shows a diploid version; the haploid version can be derived from that and is simpler). 3) Merge overlapping SVs and put them into a VCF. 4) Given a set of short-read samples, call SVs with Manta and merge short-read SVs to long-read SVs. 5) Run GraphTyper2 to genotype SVs.

Starting with accurate long-read assemblies, this pipeline can find exact SV break points and alleles. With a dozen of samples, we can ascertain many common SVs, a big fraction of which can't be found with short reads alone. More long-read assemblies are coming. This will make GraphTyper2-like approach particularly appealing. While I list the above as the first major concern, I am ok if the authors choose not to implement my proposal. I just think this manuscript will be substantially improved with proper use of long-read assemblies. See also the recent vg and paragraph preprints on graph-based SV calling (doi.org/10.1101/654566 and doi.org/10.1101/635011). I understand that these preprints just came out. I won't request comparison to them, but the authors may learn something from them.

* The 1000g NA12878 SV callset is not the gold standard these days. The authors said there are 2612 high-confidence deletions in NA12878 (on line 126). This number is too small. There should be ~10,000 \geq 50bp INDELs on a non-African haplotype relative to GRCh37/38. The authors should take either syndip [PMID:30013044] or HG002 (ftp://ftp-trace.ncbi.nlm.nih.gov/giab/ftp/data/AshkenazimTrio/analysis/NIST_SVs_Integration_v0.6/) as the ground truth. These are much better truth sets and can more comprehensively measure the SV accuracy. Another good truth set is HGSVC by Chaisson et al [PMID:30992455].

* Please provide precompiled binaries for download or better add GraphTyper to bioconda. I failed to compile GraphTyper as our machine lacks the LZ4 and the ZSTD libraries; BOOST is also too old. While I can probably resolve these dependency issues in a few hours, many common users can't. At the GraphTyper GitHub portal, 5 out of the 10 issues are related to installation. This

suggests I am not alone.

Minor comments:

- * It would be good to provide a small test dataset for users to run.
- * It would be good to discuss why starting with Manta calls, GraphTyper2 can be more accurate than Manta. This is not apparent to many readers.
- * Line 405: please explain notations (R and r)

Reviewer: Heng Li

Response to reviewers

Reviewer #1 (Remarks to the Author):

In this manuscript, the authors extend their GraphTyper tool to handle linear structural variants (insertions and deletions). This generalization is straightforward given the graphical models that they use to support their genotyping approach. The particular encoding is somewhat unusual, as instead of building a graph from the full SV allele, the authors add structural variants to the graph by adding only sequence around their breakpoints. It is also somewhat unusual in that it appears (at least based on figure 1) to be reference biased. The reference sequence is retained in the case of deletions, but the insertion sequence is not in the case of non-reference insertions. I am curious if this might affect performance for insertions and deletions. Could the authors perhaps demonstrate that their approach does not cause reference (or deletion) bias in the genotyping of indels?

Thank you for the suggestion. If our method did not have any reference bias this would make a very strong point, unfortunately, we cannot show that as our method depends on the reference sequence and clearly has some reference bias. We have now noted this in the manuscript.

In our method, we need retain the deleted reference sequence because we are interested in the graph alignment coverage inside of it. Furthermore, often the deletion overlaps some other variation that we would like to genotype. For insertions, we do not typically have access to the full SV sequence. For cases when the full SV sequence is available, we believe inserting it all would yield minor improvements at best and will often have negative impacts if the insertion is a false positive.

In the future, however, we would like to build a high quality graph containing the full SV sequences. But we note that building that graph will require a tremendous amount of work and is beyond the scope of our manuscript.

As in their previous work, their approach here is not based on a whole genome alignment of reads to a pangenome graph, but a local realignment of reads that are anchored to a particular

genomic region to a chunk of the pangenome at that locus. This limitation is not as severe as it might seem, and they showed previously that this expedient approach improves genotyping accuracy at small variants, in particular indels. (My own experience confirms this, as I developed a similar technique that was applied in the 1000 Genomes Project to yield high quality indel genotype likelihoods, but was not published outside of the supplementary material of the final project paper.)

Although its use of pangenome graphs is interesting, GraphTyper is also impressive in terms of its joint genotyping capability, and can handle tens of thousands of samples efficiently. Here, the authors build on the graph based features of GraphTyper and its joint genotyping scalability to produce a new method capable of genotyping SVs in large population cohorts. The scale that they achieve is impressive, and represents the largest combined genotyping effort for SVs that I am aware of. As such, the significance of this work cannot be questioned. However, I would like that the authors spend some more time validating their method, and I will try to provide some suggestions about other methods that they might be able to use.

To build up to their final 50k genome result, the authors first validate the approach by using a truth set established with a previous paper (svclassify) on NA12878 and comparing their performance to other SV genotyping tools (figure 2). They combine this with a study of apparent SV transmission in four large families from the Icelandic cohort, to demonstrate that the Manta+GraphTyper pipeline produces accurate SV calls that are likely to obey expected allele transmission patterns. They then examine the rate of re-call of SVs from a number of sources in the 50k sample Icelandic dataset, which provides some cross-validation as to the accuracy of the method.

To better understand the performance characteristics of their method, I suggest that the authors attempt to build and genotype a set of SVs from the "syndip" synthetic diploid benchmark (<https://www.nature.com/articles/s41592-018-0054-7>). This context provides two PacBio assemblies of fully homozygous cell lines and a lab-mixed Illumina data set combining the two lines. I suggest that the authors use SVs found by aligning these assemblies to the linear reference to build their graph, and then attempt to re-genotype them using the mixed Illumina data. This kind of evaluation would be the most generic possible given current data, and would allow the authors to describe the performance of their method for any kind of SV found between these two assemblies and the reference genome. I understand that this is a significant ask, and

must clarify that the paper in its current form presents a significant advance. I do think that the authors should consider adding another simulation study to evaluate the performance of their method where the truth can be (nearly) known for certain. This particular design, or a modulation of it (such as focusing on a subset of manually curated SVs) might be a simple way to tackle it with available data.

Thank you for the suggestion. We have followed up on it and updated the manuscript with a comparison to the syndip benchmark dataset and moved the comparison to svclassify truth set to the supplementary text. We believe your suggestion has improved our study substantially, the syndip dataset has a lot more SVs than our previous truth set.

We have also added a pipeline where we discover SVs by aligning public assemblies to the linear reference (using an approach derived from the one that was used for creating the syndip dataset). Incorporating these SVs into our pipeline has greatly improved our sensitivity, particularly of large insertions which are especially difficult to call with short reads only.

We believe simulation studies are very helpful. Internally, we routinely run simulations to test, debug and tweak our methods. Evaluating methods with simulated data can give some insights, however, we would like the manuscript to focus on real data evaluations.

I have a few minor comments about the text.

First, on line 56 of page 4, the authors describe the gVCF merging approach for joint genotyping, and note that it breaks for SVs. I would argue that there is ample evidence that this approach breaks even in the case of small indels and complex or clustered variants, due to the effects of differential allele alignment and low coverage over complex alleles. A joint genotyping approach like GraphTyper has substantial advantages in the case of complex alleles, and the authors are more than right to point out that it matters even more for SVs.

Thank you. We agree and we have now pointed that out in the manuscript.

Second, on line 59 of page 4, the authors describe pangenome graphs as extensions of the linear reference that include sequence variants. I contest this, in that I regard "pangenome graphs" as the representation of whole genome alignments of many genomes. Thus there is no particular reference genome to add variation to, but many that are equally represented. The particular case in which we have a linear reference genome that is generalized to include variation is of course useful, and equivalent to a VCF or DAG version of the pangenome. The authors' particular implementation does behave as they describe, and I think they should clarify here (as they do later) that they are working on a particular subset of the general idea as it is applied in practice, and not lay claim to the generic concept.

That is a valid point. We have now updated our definition of "pangenome graphs" accordingly. Thank you.

In their discussion, on page 12, line 204-205, they note that GraphTyper can be applied to the full structural variant spectrum. I find this slightly at odds with the fact that their method doesn't directly handle some kinds of SVs, like CNVs (amplifications), inversions, and translocations. Perhaps the authors should qualify this statement, as they do in other parts of the document.

We agree that there are still some types of structural variants that we do not handle directly. We have therefore changed the statement to "[...] genotype variants across many variant classes without any size restrictions". Thank you.

It's a minor issue that will surely be caught later, but I noticed that a paper I coauthored is listed twice in the works cited.

Thank you for noticing, this mistake has now been resolved.

I thank the authors for their excellent work on this topic,

Erik Garrison

Reviewer #2 (Remarks to the Author):

The paper proposes a new method (Graphtyper 2) to genotype structural variations (SVs) using pangenome graphs. The approach is a continuation of the authors' previous work (Graphtyper) on genotyping smaller variations (SNVs). The method consist of predicting putative SVs using a standard reference-based approach on a set of short reads on a population sample, adding start and end points (breakpoints) of these SVs to a pangenome graph consisting of SNVs that have been predicted using Graphtyper on the same data, and finally using re-aligned reads in the vicinity of breakpoints to do the final prediction of genotypes.

The method is compared against competing approaches that do not use pangenome graphs. A slight sensitivity increase is observed in experiments on a standard public dataset (100G, family trio, high-confidence deletions). This is at the cost of precision, so that the approach is competitive in F1 score only on a small range of parameter values tested (predicted breakpoint location accuracy).

While the public dataset is limited in this context, authors proceed to analysing in-house data, consisting of short and long read data on 56 and 41 Icelanders, respectively, from 4 families. The high number of offspring in the chosen family trees gives a valuable way to evaluate precision of genotype calls: Same inherited genotype should appear in more than one individual. In this experiment, the new approach appeared to provide much more sensible genotype calls than competing approach. Another analysis on family trios gave further evidence, since the transmission rate predictions of the new method were distributed around 50%, while the competing method was far from this. Long read data was used as a further evidence on the correctness of the short read calls: 67.7% of calls were supported by long reads.

As a summary, while the improvements on public dataset are rather modest, the in-house experiments complement well. As far as I know, this kind of analysis on large family cohort is novel. I find this analysis a bigger contribution than the proposed method itself, which consist of somewhat ad hoc steps like greedy clustering of close-by breakpoints, huge number of fixed parameters (breakpoint size, coverage cutoffs, etc.). Some of these are unavoidable, but I would appreciate more rigorous approaches to the subproblems involved for repeatability and adjustability to different situations.

Our main focus was on developing a method that could scale to a large number of individuals in a joint analysis. We agree that our method has many steps with arbitrarily selected parameters and thresholds, and those should be optimized in the future. However, we believe our method is a substantial contribution and a practical solution for jointly genotyping SNPs, indels, and SVs in ~50k whole-genome sequenced individuals.

We have implemented a proposal by reviewer #3 such that SVs in public long-read assembly data would be incorporated into our pipeline. We have also changed our public truth set to a more complete dataset (syndip). With these changes, we see that our method offers even further improvements.

The success on in-house data raises some concerns. Why was only Manta used as a comparison? Why not Manta+BayesTyper, Delly, Lumpy+SVTyper, as with the public dataset?

Thank you. In this experiment, we wanted to demonstrate that GraphTyper greatly improves the genotype accuracy of Manta SVs compared to the original calls. This is an important point to make because it showed us that applying graph alignments to Manta SVs has advantages over just merging the original calls. However, we can understand that a comparison to other methods can put the results of Manta and Manta+GraphTyper into perspective. Therefore, we have added a comparison to Delly and smooove (which essentially runs Lumpy+SVTyper along with a few other tools).

Authors provide an implementation of Graphtyper2. The software works as expected: Can be installed easily and runs without problems on small dataset. On data on a complete chromosome a standard desktop is not sufficient, so such analyses need to be run on servers equipped with sufficient main memory.

Some more specific comments follow.

Authors do not formally define a breakpoint, but introduce an implicit definition in lines 87-88, which is the intuitive meaning of breakpoint. However, the word breakpoint and breakpoint

sequence are used in ways that appear to mean something else, without much clarification. Figure 1.a and Supplementary Figure 2 are supposed to help, but they add little or no insight.

Some excerpts:

87-88: "Two breakpoints are typically added for each SV, representing the start and end locations of the SV (...)"

92-93: "GraphTyper inserts only the breakpoint sequences (up to 152 bp (...))"

So, if breakpoints are locations, what exactly are breakpoint sequences ?

Moreover lines 111-113 look almost like a circular definition:

111-113: "Briefly, breakpoints are genotyped by counting the number of reads aligning through the different paths of a breakpoint. Each breakpoint is genotyped separately for variants with two breakpoints."

Figure 1.a should clarify this, but it doesn't help much.

One could hope that the method sections would provide more precise definition, but that was not the case, as in section "Relative genotype likelihoods" we still see some ambiguity, e.g in

390 "(...) comparing how many reads support a given SV breakpoint compared to the reference allele" we get the impression that the SV breakpoint is actually the actual SV allele. However, immediately after:

391-392 "We consider a read to support an allele if its best graph alignment is in a path that overlaps the allele"

Again, supplementary Figure does not add much insight here.

Now, after line 399 "SV breakpoint" appears to mean the actual breakpoint as originally stated.

Thank you for pointing this out. We agree that we did not explicitly define these terms and we used them inconsistently a few times.

We have now updated the manuscript such that it defines both breakpoint and breakpoint sequence explicitly. We have also updated Figure 1a such that it now has a label next to the breakpoint sequence for further clarification.

Then a second comment (minor) about the experiments;

188-192 "As before, we merged SVs discovered by Manta which resulted in 543,939 SVs discovered in the population. These SVs were added to the SNP and indel graphs that were previously created with GraphTyper. Subsequently, we genotyped all samples using GraphTyper which resulted in 486,158 SVs that were called in at least one sample."

What is the reason that the result after using GraphTyper is a reduced number of SVs ? (maybe it is obvious, but still a comment will be welcomed).

Moreover, this would appear to be at odds with Figure 2 (although it is a different dataset), in which Manta+GraphTyper has almost the same precision as Manta alone, and the actual improvement of using Manta+GraphTyper (against just Manta) is in the sensitivity.

The number of SVs is reduced because some of the discovered SVs were not genotyped in any sample with GraphTyper, i.e. there are variants identified by Manta and then not confirmed by GraphTyper. We have added a clarification of this in the manuscript.

In Figure 2 (which has now been moved to the supplementary text), we are only evaluating the manta SVs that passed Manta filter (have "PASS" in the VCF). When we merge SVs we include all Manta SVs, this is why it is possible for Manta+GraphTyper to have greater recall than Manta alone. We could evaluate all Manta SVs to increase their recall slightly, but Manta's overall performance would be lower since its false discovery rate would be substantially increased.

We believe with our analysis of SVs in the four Icelandic families we are showing that the Manta+GraphTyper pipeline is not only improving the sensitivity but also the

genotype accuracy of the calls. We both have a much lower Mendelian inheritance error and in our parent-offspring analysis, we see that the SVs are transmitting to the offspring at their expected rate.

Last, the merging of nearby breakpoints appears a bit ad hoc. Have you considered using a dynamic programming approach for segmenting the breakpoint locations? Or using some clustering tool that can also take the breakpoint sizes into account (defining a distance between two breakpoints as a function of distance and size difference)?

Thank you, we do agree with the reviewer that this method is ad hoc and not optimal. We believe that this is an important problem and have pointed this out as an open problem in the discussion. We have dabbled with using other more principled methods and believe that the problem is quite complicated to solve properly and to do so requires insights into a) Evolutionary processes that created the variants and b) The error models of the algorithms that identified the variants. This finally needs to be merged with efficient and correct algorithms. We will consider these options in the future.

Reviewer #3 (Remarks to the Author):

Eggertsson et al described GraphTyper2, a graph-based short-read genotyper for both small variants and structural variations (SVs). It is the successor of GraphTyper published in Nature Genetics two years ago. The major improvement in this new version is the ability to call SVs.

GraphTyper2 takes a list of potential SVs, incorporates them into a sequence graph and genotypes them in a set of samples. In the manuscript, the authors ran GraphTyper2 on SV calls generated by the Manta SV caller. To some extent, GraphTyper2 refines Manta calls, takes the advantage of multi-sample information and achieves higher accuracy, as is shown in authors' evaluation. The idea behind GraphTyper2 is sound and the algorithm is relatively straightforward. I believe it can be applied to a variety of large cohorts for SV calling which is currently missing in most studies.

Critical observations are as follows:

* In the manuscript, the authors generated SV candidates with Manta from short reads. Its sensitivity is largely limited by short reads. On the other hand, I see the key advantage of genotyping is that we can use SV candidates found by other technologies. I believe the authors can achieve much higher sensitivity by incorporating long-read SVs into the graph.

Here is a proposal. 1) Download a dozen or so long-read assemblies described in PMID:30661756. 2) Map these assemblies to the human reference genome using the pipeline described here: <https://github.com/lh3/CHM-eval/tree/master/dip-call> (the page shows a diploid version; the haploid version can be derived from that and is simpler). 3) Merge overlapping SVs and put them into a VCF. 4) Given a set of short-read samples, call SVs with Manta and merge short-read SVs to long-read SVs. 5) Run GraphTyper2 to genotype SVs.

Starting with accurate long-read assemblies, this pipeline can find exact SV break points and alleles. With a dozen of samples, we can ascertain many common SVs, a big fraction of which can't be found with short reads alone. More long-read assemblies are coming. This will make GraphTyper2-like approach particularly appealing. While I list the above as the first major concern, I am ok if the authors choose not to implement my proposal. I just think this manuscript

will be substantially improved with proper use of long-read assemblies. See also the recent vg and paragraph preprints on graph-based SV calling (doi.org/10.1101/654566 and doi.org/10.1101/635011). I understand that these preprints just came out. I won't request comparison to them, but the authors may learn something from them.

Thank you for your proposal. We have implemented your proposal and we believe it has substantially improved our manuscript. We measured an increase in recall across all SV sizes by discovering SVs from these public assemblies, in particular, we saw large improvements in genotyping large insertions (2000 bp+) where we observed approximately 3-fold increase in recall.

We have added citations to these preprints.

* The 1000g NA12878 SV callset is not the gold standard these days. The authors said there are 2612 high-confidence deletions in NA12878 (on line 126). This number is too small. There should be ~10,000 \geq 50bp INDELS on a non-African haplotype relative to GRCh37/38. The authors should take either syndip [PMID:30013044] or HG002 (ftp://ftp-trace.ncbi.nlm.nih.gov/giab/ftp/data/AshkenazimTrio/analysis/NIST_SVs_Integration_v0.6/) as the ground truth. These are much better truth sets and can more comprehensively measure the SV accuracy. Another good truth set is HGSVC by Chaisson et al [PMID:30992455].

Thank you for the suggestion. We followed up on it and have now updated the manuscript with a comparison to the syndip benchmark dataset and moved the comparison to svclassify truth set to the supplementary text. We believe your suggestion has improved our study greatly, the syndip dataset captures a lot more SVs than svclassify.

* Please provide precompiled binaries for download or better add GraphTyper to bioconda. I failed to compile GraphTyper as our machine lacks the LZ4 and the ZSTD libraries; BOOST is also too old. While I can probably resolve these dependency issues in a few hours, many common users can't. At the GraphTyper GitHub portal, 5 out of the 10 issues are related to installation. This suggests I am not alone.

Thank you. We have added precompiled binaries for download on GraphTyper's github page. This should make installation much easier. We are also considering the possibility of adding GraphTyper on bioconda.

Minor comments:

* It would be good to provide a small test dataset for users to run.

We have added a small test dataset at https://github.com/DecodeGenetics/graphtyper-pipelines/tree/master/test_sv

* It would be good to discuss why starting with Manta calls, GraphTyper2 can be more accurate than Manta. This is not apparent to many readers.

We have added a clarification why the Manta+GraphTyper pipeline can have higher recall than the original Manta calls.

* Line 405: please explain notations (R and r)

Thank you. We have added an explanation of these notations.

Reviewers' comments:

Reviewer #1 (Remarks to the Author):

The authors have made a very reasonable response to all of my concerns. The addition of the simulation study makes a very convincing case as to the performance of the method. I find the work to be sound, and look forward to applying it for SV calling in large cohorts of genomes.

-Erik Garrison

Reviewer #2 (Remarks to the Author):

Authors have adequately responded to my concerns. The new experiments added, following suggestions by other two reviewers, also make the paper stronger.

Reviewer #3 (Remarks to the Author):

In this revision, the authors made two major revisions: changing the SV truth set to syndip, and using SVs from long-read assemblies as a source of candidate SVs. These two changes addressed most part of my major concerns in the previous round of review. I only have one major comment on the new materials.

* It worries me that in Figure 2b, the FDR (false discovery rate) of Manta+AA+GraphTyper can be as high as 40% (i.e. 40% of deletion calls are wrong). This also makes me realize that the manuscript may be missing one more evaluation.

Note that in single-sample evaluation, GraphTyper2 can assume all candidate SV calls come from the sample. With this assumption, GraphTyper2 can pick up SV candidates with traces of evidence. However, when we include SVs from other samples, many candidates are absent from the target sample. The low FDR of Manta+GraphTyper2 on a single syndip sample doesn't necessarily indicate low FDR on population calling. On line 224, the authors said that 67.7% of SV calls from the 41 Icelanders are validated by Sniffles. This again implies high FDR.

Given the observation above, I would like to see one additional call set for syndip. The authors can call SV candidates from the 41 Icelanders and syndip, and then use them to genotype syndip. In particular, I wonder if FDR is increased with candidate SVs from other samples. Relatedly, the authors may use more long-read assemblies from the MGI reference project (https://urldefense.proofpoint.com/v2/url?u=https-3A__www.genome.wustl.edu_items_reference-2Dgenome-2Dimprovement_&d=DwIFAg&c=vh6FgFnduejNhPPD0fl_yRaSfZy8CWbWnIf4XJhSqx8&r=2oJvShlC5KdacAoel-2Y9y34fAnKxayvMsk4b8qyM5g&m=qWaFZisKni57oOqp3rxDnhdgRloX8DY3rVvQfBddHEc&s=LFqP0uChjLbi3U19Yhyfdjvnmn7D1b2C2I15pYub-VmA&e=). I would be worried if the FDR of Manta+AA+GraphTyper continues to climb with more samples. In my view, FDR should not be greatly affected by candidates absent from the target sample. If GraphTyper2 is indeed affected, the authors may consider to distinguish SV candidates called from the target sample versus SV candidates called from other samples, for example, by setting different thresholds.

I suggest the authors look into false positive calls to understand why the FDR of Manta+AA+GraphTyper is so high. If they can convince me that high FDR in Figure 2b is inevitable, they may not need to generate the additional call set as is described in the last paragraph.

Reviewer: Heng Li

Response to reviewers

Reviewer #1 (Remarks to the Author):

The authors have made a very reasonable response to all of my concerns. The addition of the simulation study makes a very convincing case as to the performance of the method. I find the work to be sound, and look forward to applying it for SV calling in large cohorts of genomes.

-Erik Garrison

Thank you for your comments and suggestions.

Reviewer #2 (Remarks to the Author):

Authors have adequately responded to my concerns. The new experiments added, following suggestions by other two reviewers, also make the paper stronger.

Thank you for your comments and suggestions.

Reviewer #3 (Remarks to the Author):

In this revision, the authors made two major revisions: changing the SV truth set to syndip, and using SVs from long-read assemblies as a source of candidate SVs. These two changes addressed most part of my major concerns in the previous round of review. I only have one major comment on the new materials.

* It worries me that in Figure 2b, the FDR (false discovery rate) of Manta+AA+GraphTyper can be as high as 40% (i.e. 40% of deletion calls are wrong). This also makes me realize that the manuscript may be missing one more evaluation.

Note that in single-sample evaluation, GraphTyper2 can assume all candidate SV calls come from the sample. With this assumption, GraphTyper2 can pick up SV candidates with traces of evidence. However, when we include SVs from other samples, many candidates are absent from the target sample. The low FDR of Manta+GraphTyper2 on a single syndip sample doesn't necessarily indicate low FDR on population calling. On line 224, the authors said that 67.7% of SV calls from the 41 Icelanders are validated by Sniffles. This again implies high FDR.

Thank you. You are correct that GraphTyper2 can pick up SV candidates with traces of evidence. This is an intended behavior as the method attempts to compare evidence for both reference and the candidate alt. alleles to genotype them as nearly equivalent alleles (there is some reference bias as we note in the discussion). We believe that by doing that we increase our recall since true variants can have low evidence. Thus, we report variants (SNPs, indels and SVs) even with low evidence. However, we provide multiple different quality metrics to filter on for scenarios where low FDR is critical.

In Supplementary Note Figure 3a we showed that GraphTyper needed 3x avg. coverage to recall ~80% of the svclassify deletions in the NA12878 individual. Due to the low coverage, most of these calls were made with very low evidence. GraphTyper's SNP+indel genotyping was also designed in a similar manner. We believe this makes the method more versatile as we can filter the results differently based on the requirements we have at hand. Of course we try to minimize false positives, so we have in place very lenient filters to only remove variants that have nearly zero chance of being real germline variants.

We agree that the ~40% FDR of Manta+AA+GraphTyper is fairly high compared to the ~20% FDR of Manta+GraphTyper. This may be problematic in some scenarios, but we suggest solving that in post-filtering using the variant quality metrics. The FT field in the genotype call is a simple and effective filter. For example, by removing deletions with non-PASS FT field, we take out nearly 90% of the false positives, however, at the same time we lose about 40% of the true positive deletions (both values measured at a 50 bp breakpoint precision threshold). Since we see that a lot of the TP deletions have low evidence in the short reads, we do not want to enforce the filter by default. Further, if the sample had very low coverage, the filter would likely remove all or almost all of the calls. Finally, rather than filtering on the short read evidence, we often rather want to filter the

data based on other orthogonal data, such as (1) inheritance in parent-offspring trios (2) population imputation quality scores, and (3) various genome annotations like mappability and tandem repeats.

Thank you, we have added some discussion on this point to our manuscript.

Given the observation above, I would like to see one additional call set for syndip. The authors can call SV candidates from the 41 Icelanders and syndip, and then use them to genotype syndip. In particular, I wonder if FDR is increased with candidate SVs from other samples. Relatedly, the authors may use more long-read assemblies from the MGI reference project (https://urldefense.proofpoint.com/v2/url?u=https-3A_www.genome.wustl.edu_items_reference-2Dgenome-2Dimprovement_&d=DwIFAg&c=vh6FgFnduejNhPPD0fl_yRaSfZy8CWbWnlf4XJhSqx8&r=2oJvShlC5KdacAoel-2Y9y34fAnKxayvMsk4b8qyM5g&m=qWaFZisKni57oOqp3rxDnhdgRloX8DY3rVvQfBddHEc&s=LFqP0uChjLBi3U19Yhyfdjvmn7D1b2C2I15pYub-VmA&e=). I would be worried if the FDR of Manta+AA+GraphTyper continues to climb with more samples. In my view, FDR should not be greatly affected by candidates absent from the target sample. If GraphTyper2 is indeed affected, the authors may consider to distinguish SV candidates called from the target sample versus SV candidates called from other samples, for example, by setting different thresholds.

Thank you for providing us with more assemblies to use.

We agree that it would make a strong point to run the proposed experiment if we could show that FDR in one sample did not increase when adding more SV candidates from other individuals. Unfortunately, we cannot show this and believe it is a common problem among joint calling methods.

We are worried that the proposed filtering solution to the problem would bias our genotype calls towards Manta genotypes. This is problematic, since Manta is greatly underestimating the frequencies of SVs (Figure 3b-d), i.e. often a common SV is only detected in one or a few individuals in the population. We suggest filtering on read evidence rather than the discovery method output.

I suggest the authors look into false positive calls to understand why the FDR of Manta+AA+GraphTyper is so high. If they can convince me that high FDR in Figure 2b is inevitable, they may not need to generate the additional call set as is described in the last paragraph.

We manually inspected some of the false positive Manta+AA+GraphTyper deletions that had a call with "PASS", i.e. they had high evidence in the short-read data but still were considered a false positive. Many of these deletions are represented somewhat differently than in the truth set. We note that there is no SV discovery in GraphTyper, it only considers the reference and alt. alleles as possibilities at a given locus, meaning it would call a slightly incorrect alt. allele if it was "closer" to the correct variation than the reference.

For example, in 37d5 there are two deletions on the same haplotype at genome coordinates 1:755,374-755,829 (455 bp) and 1:755,850-756,519 (669 bp). They have only 21 bp between them and in total 1124 bp are deleted. GraphTyper genotyped a deletion at 1:755,402-756,522 (1120 bp) which is considered a false positive. GraphTyper indicated that this deletion has a high quality score (QUAL=208), which is understandable since it gathered evidence from these two similar SVs. Assuming that the deletion is really two separate events with 21 bp between them, it would be very difficult to recognize that in the short-read data. The 1120 bp deletion allele is much closer to the truth than the reference allele and therefore it was called in GraphTyper. GraphTyper was the only evaluated genotyping method which called any deletion at a nearby site in CHM1/CHM13. In the 1000G genomes SV publication (Sudmant et al., 2015) there is a 1447 bp deletion (single event) in the locus. It is difficult to determine whether all of these deletions should be the same event and which of these representations are correct (if any).

Then we also see some cases of deletions that have high evidence in the short read data, but with no similar SVs in the truth set. For instance, the deletion at genome coordinates 1:144,896,297-144,901,046 (4749 bp), which was genotyped only in Manta and GraphTyper. We can see a clear drop of alignment coverage in the region and there are multiple read pairs with abnormal insert sizes that span the region. In some other occasions all evaluated methods called a deletion that is not present in the truth set, for

example, the deletion at 1:155,844,263-155,845,011 (748 bp) which has high evidence indicated by split aligned reads, alignment coverage, and spanning read pairs. It is surprising to see deletions with that much evidence being a false positive. We suspect that some of these cases are false negatives in the truth set.

In summary, the 40% deletion FDR is not entirely inevitable and can be heavily reduced by applying simple filters. Some of the remaining false positives have very strong evidence and would be difficult to remove without taking out many true positives as well. We will continue to work on improvements in the method to reduce the false positives, some of those are noted in the discussion.

Reviewer: Heng Li

Thank you for your comments and suggestions. We believe they have substantially strengthened our study.

REVIEWERS' COMMENTS:

Reviewer #3 (Remarks to the Author):

The authors' explanation to high FDR is satisfactory.